# Eating Behaviors during Pregnancy: A Cross-Sectional Study

**DOI:** 10.3390/healthcare12161616

**Published:** 2024-08-13

**Authors:** Jawaher Al Hamimi, Asma Al Shidhani, Maya Al Mamari, Ahmed Al Wahaibi, Salah T. Al Awaidy

**Affiliations:** 1Department of Family Medicine, Directorate General of Khoula Hospital, Ministry of Health, Muscat 113, Oman; jawaheralhamimi@gmail.com (J.A.H.); m.memo20200@gmail.com (M.A.M.); 2Department of Family Medicine, Sultan Qaboos University Hospital, Muscat 123, Oman; asma.shidhani@hotmail.com; 3Directorate of Primary Health Care, Ministry of Health, Muscat 113, Oman; ahmedalwahaibi@hotmail.com; 4Freelance Public Health Consultant, Muscat 111, Oman

**Keywords:** eating behaviors, pregnant women, socio-demographic factors, Oman

## Abstract

Background: Eating disorders are complex illnesses with serious long-term consequences. They are linked to negative outcomes such as miscarriage, low birth weight, and other obstetric and postpartum difficulties. Our study in Muscat, Oman, examines the eating habits of pregnant women who consult primary care physicians. In this study, our aim is to identify key sociodemographic factors linked with eating disorders in Oman. Method: We used the Eating Disorder Examination Questionnaire to assess the potential for the presence of eating disorders. Chi-Square and Fisher’s Exact tests were used to analyze relationships between improper eating behavior and independent variables. Results: The study showed that 1.6% of participants had a potential diagnosis of an eating disorder, with the most common inappropriate behavior being binge eating at a prevalence of 18.8%. A pre-gestational low Body Mass Index (BMI) was associated with a higher prevalence of binge eating during pregnancy. Additionally, we found that pregnant women who were working were more prone to self-induced vomiting. High BMI before pregnancy was significantly associated with various inappropriate eating behaviors, such as restraint behavior (*p* = 0.000), shape concern (*p* = 0.000), weight concern (*p* = 0.040), eating (*p* = 0.045), laxative use (*p* = 0.020), and excessive exercise (*p* = 0.043). Conclusion: The study reveals a high prevalence of eating disorders in pregnancy. Less educated women exhibit higher laxative use, while working women show more instances of binge eating and self-induced vomiting. These findings emphasize the critical need to prioritize targeted interventions and support for vulnerable pregnant women.

## 1. Introduction

Eating disorders are complex conditions that can harm physical and mental health and can persist over time, with periods of improvement and relapse [1,2]. Although the causes of these disorders remain incompletely understood, a combination of genetic, psychological, and sociocultural factors are believed to be responsible. The main types of eating disorders are anorexia nervosa, bulimia nervosa, and binge-eating disorder. Anorexia is characterized by a fear of weight gain and a distorted body image, resulting in low body mass. Binge-eating disorder involves overeating with a loss of control, while bulimia involves compensatory behaviors like vomiting or excessive exercise [1]. Pregnancy involves significant changes to the body, and body dissatisfaction can impact pregnant women with or without an eating disorder. As a result, eating disorder symptoms may arise or worsen during pregnancy [3,4].

Pregnant women may still feel societal pressure to maintain a slim body or struggle with obesity. Excessive weight gain during pregnancy can lead to postpartum weight retention and increase the risk of developing obesity [5]. Binge eating is a common form of eating disorder during pregnancy and can cause significant weight gain. Studies showed that 25–44% of women experience regular binge eating during their first pregnancy [3,6,7,8]. Women who have a history of restrictive eating patterns or overeating in response to negative emotions are more likely to experience binge eating during pregnancy. Moreover, some women may struggle to determine what constitutes appropriate weight gain during pregnancy and may have distorted perceptions of their body image [9].

During the second and third trimesters of pregnancy, increased energy needs support fetal growth, maternal tissue expansion, and heightened metabolic demands. Overeating can lead to excessive weight gain, raising risks such as gestational diabetes, preeclampsia, and delivery complications. Conversely, inadequate food intake in eating disorders can deprive essential nutrients, potentially causing insufficient fetal growth, low birth weight, and developmental challenges due to maternal malnutrition. Metabolic disorders like insulin resistance and dyslipidemia linked to these disorders contribute to complications such as gestational diabetes and cardiovascular issues, posing difficulties in weight management for both mother and baby. Eating disorders during pregnancy also increase the risks of miscarriage, premature delivery, and developmental delays in children [10]. Additionally, women with eating disorders may face challenges with breastfeeding and bonding with their newborns [10]. The treatment of eating disorders during pregnancy necessitates a comprehensive, multidisciplinary approach involving medical, nutritional, and mental healthcare [10,11].

Although eating disorders have negative effects on both maternal and fetal health, there is currently no recommended tool for identifying these disorders in pregnant women. To address this gap, the present study utilizes the Eating Disorder Examination Questionnaire (EDE-Q), a widely recognized instrument for assessing and diagnosing eating disorders in non-pregnant adults [12], to measure the prevalence of eating disorders in pregnant women.

There is a dearth of research on eating disorders during pregnancy in Oman and the larger Gulf Cooperation Council (GCC) countries. Furthermore, there is limited knowledge about how women adapt to the changes in their body weight and shape during pregnancy, particularly among those who reported inappropriate eating habits before becoming pregnant. As a result, there is a lack of understanding about the emergence of these behaviors during pregnancy. Thus, in this study, we aim to identify significant sociodemographic factors associated with eating disorders in Oman.

## 2. Methods

### 2.1. Study Design

In January and April 2022, we conducted a cross-sectional study among Omani pregnant women who attended antenatal clinics in 15 local health centers and 3 polyclinics in the Muscat governorate.

### 2.2. Sample Size and Study Population

The prevalence of possible eating disorder among pregnant women, according to the literature, varies considerably across studies, ranging from 0.6% to 27.8% [3,5,6]. We calculated a sample size of 700 participants based on an 18% prevalence with a precision of 3%. We selected pregnant women who were 18 years old and older, in their second and third trimesters, and attending primary healthcare centers using nonprobability convenient sampling. The study excluded non-Oman citizens from participation.

### 2.3. Data Collection Tool, Data Collection and Procedures

We translated a questionnaire into Arabic and conducted a pilot study on a smaller group of 15 pregnant women to assess their comprehension of written text. Competent general practitioners administered the tool effectively. We enlisted general practitioners in primary healthcare and provided them with instructions regarding the study’s aims and the data gathering process. We gathered the data through in-person (face-to-face) interviews conducted in Arabic, using a customized structured questionnaire. We obtained the informed consent of the participants in advance. We guaranteed the confidentiality of the gathered information to the participants, and their participation in the research was entirely voluntary. No financial compensation was provided.

We used the Eating Disorder Examination Questionnaire (EDE-Q), a validated questionnaire, to collect sociodemographic data such as age, as well as obstetric information, including parity, number of children, planned pregnancy status, single or multiple pregnancy, and gestational diabetes mellitus (GDM). Social variables, including educational level (ranging from illiteracy to completion of college or higher education), employment status (employed, homemaker, student, or retired), family structure (nuclear or extended), household income (classified as low ≤ 350 Oman Riyals (OMR), moderate 500–1000 OMR, high > 1000 OMR), smoking status, intention of pregnancy, history of prior abortions, as well as the presence of additional medical conditions and concurrent mental health disorders, were also collected. During the visit, we took anthropometric measurements such as weight and height to calculate the body mass index (BMI) in kg/m^2^, and calculated the gestational age based on the last menstrual period (LMP).

The study examined eating behaviors using the self-reporting EDE-Q. Copyright protects the EDE-Q (and its items). You can freely use it for non-commercial research purposes without seeking permission. The EDE-Q questionnaire comprises a total of 28 items, with 22 specifically designed to assess the core symptoms of eating disorders (EDs). We further divide these 22 items into four main areas or subscales, which include dietary restraint (five items), eating concern (five items), weight concern (five items), and shape concern (eight items) over the past 28 days. We evaluate the frequency or intensity of each item using a seven-point Likert scale, which ranges from 0 (absence of the symptom) to 6 (presence of the symptom every day or to an extreme degree). We sum and average the items within each subscale to determine the subscale scores. We use the sum of subscale scores to determine the global score, which reflects the severity of ED symptoms. A higher score on the questionnaire indicates greater ED symptomatology.

### 2.4. Statistical Analysis

Categorical and numerical variables, prevalence frequencies, and percentages were reported. To determine the associations between inappropriate eating behavior and the independent variables, Chi-Square and Fisher’s Exact tests were used. Significant values were considered for *p* < 0.05. We performed the statistical analyses using both SPSS v.23 and Stata v.9.2 software.

## 3. Results

We distributed a total of 1000 questionnaires to pregnant women who attended ANC care. Out of these, 712 completed questionnaires (a response rate of 71%). However, the study excluded 78 women because they did not complete the questionnaire, were in the first trimester of pregnancy, or were not Omani citizens. Ultimately, we used the responses of 634 participants who met the inclusion criteria for data analysis.

The mean age of the participants was 30.75 (SD: 5.18), with almost two-thirds of the participants between 26 and 35 years of age (26–32%, *n* = 207; 31–35%, *n* = 192). Most women (77.4%, *n* = 419) had no history of previous abortions. A significant proportion of the sample (82%, *n* = 520) reported having a good family income, while more than half (52.4%, *n* = 277) were unemployed (Table 1). Fifty-five per of the participants, 54.9% (*n* = 348) were in the second trimester of their pregnancy, with 17.5% (*n* = 106) classified as obese and 4.5% (*n* = 27) as underweight. Thirty-seven percent of participants (*n* = 273) had a history of gestational diabetes mellitus (GDM).

### Prevalence of Eating Disorders and Inappropriate Eating Behaviors

A total of 1.6% of the women had a probable diagnosis of an eating disorder during pregnancy (Table 2). The most common symptoms of eating disorders observed during pregnancy were binge-eating episodes (reported by 18.8% of participants) and self-induced vomiting (reported by 14.4% of participants). None of the participants reported misusing diuretics, and only eleven participants reported engaging in excessive exercise during pregnancy.

Inappropriate eating behaviors such as restraint behavior (*p* = 0.000), shape concern (*p* = 0.000), weight concern (*p* = 0.040), eating (*p* = 0.045), laxative use (*p* = 0.020), and excessive exercise (*p* = 0.043) were associated with a high BMI during pregnancy. Furthermore, we found that women with a higher school level of education were more likely to use laxatives (*p* = 0.027, 60%). On the other hand, binge eating and self-induced vomiting were higher in working women (*p* = 0.031 and *p* = 0.03, respectively). In addition, all binge-eating cases were in the third trimester (*p* = 0.041). As we did not perform bivariate and multivariate analyses, we also do not know the direction of the association.

## 4. Discussion

In this study, we report a prevalence of 1.6% of a potential diagnosis of an eating disorder, with the most common inappropriate behavior being binge eating at a prevalence of 18.8%, followed by self-induced vomiting at a prevalence of 14.4%. There is a range of prevalence rates for eating disorders in pregnant women across different studies [2,3,7,13,14,15]. Using DSM-IV criteria, a recent study in southeast London in 2020 calculated the weighted prevalence of lifetime ED in a pregnant woman as 15.35% and the current ED as 1.47% [14]. In London in 1999, a previous study estimated that 4.9% of women attending follow-up antenatal clinic appointments scored above the clinical threshold on the Eating Attitudes Test [15]. In another study in Brazil in 2009, 0.6% of pregnant women indicated a probable diagnosis of eating disorder using the Eating Disorder Examination Questionnaire [3]. Using the Eating Disorder Diagnostic Scale, a 2009 study in the United Kingdom discovered that 7.5% of pregnant women met the diagnostic criteria for an ED [7]. Various factors can account for the differences in prevalence rates among studies investigating eating disorders in pregnant women. The stage of pregnancy, the specific type of disordered eating behavior under study, the assessment tool for measuring eating disorder symptoms, and the threshold for identifying clinically significant scores are among these factors. These variations may contribute to the discrepancies in prevalence observed across studies, including our own.

Binge-eating behavior was the most reported behavior, with rates ranging from 8.8% to 17.3% [3,7]. Several factors contribute to the high prevalence of binge-eating disorder (BED) during pregnancy, which our study found to be 18.8%. Firstly, the hormonal changes and physiological adaptations that occur during pregnancy can lead to increased appetite and cravings, which may contribute to the episodes of excessive eating characteristic of BED. In Oman, similar to many other regions, pregnant women may experience a range of hormonal changes, physiological adaptations, and emotional challenges that could potentially influence the prevalence of binge eating disorder (BED) during pregnancy. Moreover, the emotional and psychological changes experienced during pregnancy, such as increased stress, anxiety, and concerns about body image, may also contribute to the development or exacerbation of binge-eating behaviors. In Oman, societal expectations and cultural norms surrounding pregnancy, along with the pressures of balancing traditional roles and modern expectations, could further amplify these emotional stressors.

The current study found that a high pre-gestational BMI was associated with a high total score for possible eating disorders and binge eating during pregnancy. This stands in contrast to a previous Brazilian study that linked a pre-gestational low BMI to a higher prevalence of binge eating during pregnancy. Additionally, our study found that pregnant women who were working were more prone to self-induced vomiting, possibly due to stressors associated with their jobs. Surprisingly, age was not found to be significantly associated with a potential diagnosis of an eating disorder in this study, even though most of the sample was young (26–30 years old), and this is the age at which eating disorders are typically diagnosed. We conducted a 7-month survey to understand the experiences of expectant and postpartum women with a history of eating disorders, in contrast to our reported findings. We conducted the survey via Netmums, a national parenting website based in the UK with over 1.7 million members.

In our study, all cases of disordered eating behaviors occurred exclusively in the third trimester. This phenomenon may be attributed to several factors specific to this stage of pregnancy. Physiological changes during the third trimester, such as increased hormonal fluctuations and the physical discomfort associated with late-stage pregnancy, can exacerbate pre-existing eating disorders or trigger new onset behaviors. Additionally, the heightened emotional stress and anxiety commonly experienced as childbirth approaches may further contribute to the onset or exacerbation of disordered eating behaviors during this period.

The survey showed that younger age, a previous eating disorder, a low educational level, and previous miscarriage were significantly associated with the development of disordered eating behavior in pregnant mothers [15]. One possible explanation for the lack of age-related significance in our study could be the relatively narrow age range of our sample (26–30 years old). It is plausible that a broader age range, encompassing adolescents or older women, might reveal different patterns. Furthermore, societal shifts in body image ideals, as well as increased awareness and acceptance of eating disorders across age groups, could influence these dynamics. Moreover, the absence of a significant association between age and eating disorders in our study raises questions about the complex interplay of various risk factors. While age may play a role, it seems to interact with other variables such as socioeconomic status, psychological well-being, and prior medical history.

The generalizability of our findings may be limited due to the specific demographic and geographic context of our sample. Factors such as regional cultural norms, socioeconomic disparities, and healthcare access within our studied population could influence the prevalence and manifestation of eating disorders differently compared to other regions or populations with distinct demographic profiles and societal contexts.

### 4.1. Study Limitations

Our study has some limitations that need to be considered. Firstly, we collected data on eating behavior retrospectively, potentially leading to recall bias. Another potential limitation is the use of screening tools to identify possible eating disorder symptoms rather than structured interviews. This approach could lead to false-positive results, as screening tools may not always accurately identify eating disorder symptoms in all individuals. In addition, the use of screening tools may not provide a comprehensive understanding of the complexity and severity of eating disorder symptoms in pregnant women. Therefore, to further investigate the prevalence and impact of eating disorders during pregnancy, we may need to conduct future studies that utilize structured interviews and assessments. Furthermore, our study did not assess for the presence of other psychological disorders, such as depression and anxiety, which may be associated with eating disorder-like behaviors.

### 4.2. Study Strength

The strengths of the study include the utilization of a well-validated and widely accepted tool, such as the EDE-Q, which ensures the accuracy and consistency of the measurements. Additionally, a high response rate reduces the risk of non-response bias and enhances the representativeness of the sample. In order to facilitate follow-up and provide any necessary assistance, pregnant women who were identified as potentially having an eating disorder in our study were provided with feedback, and the authorities were notified.

## 5. Conclusions and the Way Forward

The study identifies a high prevalence (1.6%) of probable eating disorders during pregnancy. A high pre-pregnancy BMI is significantly associated with various inappropriate eating behaviors. Laxative use is higher in less educated women, while eating and self-induced vomiting are more common in working women. Notably, all cases of eating disorder occur in the third trimester. Future research could delve deeper into these interactions to elucidate the nuanced factors contributing to the development of eating disorders during pregnancy. Longitudinal studies tracking women from preconception through pregnancy and postpartum could provide valuable insights into the temporal dynamics of these disorders, as well as help inform targeted interventions and support strategies for expectant mothers at risk.

## Figures and Tables

**Table 1 healthcare-12-01616-t001:** Sociodemographic characteristics of the sample.

Characteristics		Total *n* = 634	%
Age			
	<25 Years	114	18
	26–30 Years	207	32
	31–35 Years	192	30
	>35 Years	121	19
Body mass index (BMI) kg/m^2^ (before pregnancy)			
	Underweight	28	4.4
	Normal weight	312	29.2
	Overweight	173	27.3
Gravidity			
	<3	522	82.3
	4–9	108	17
	≥10	4	0.6
Gestational Age			
	Second trimester	348	54.9
	Third trimester	286	45.1
Previous abortion			
	Yes	143	22.6
	No	419	77.4
Planed pregnancy			
	Yes	422	66.6
	No	211	33.3
Gestational diabetes mellitus (GDM)			
	Yes	273	37.4
	No	397	62.6
Level of education			
	Illiteracy	7	1.1
	School level	205	32.3
	College and above	419	66.1
Family structure			
	Nuclear family	493	77.8
	Extended family	141	22.2
Economic level			
	Low ≤350 OMR	32	5
	Good 500–1000 OMR	520	82
	High >1000 OMR	82	12.8
Smoking			
	Yes	10	1.6
	No	624	98.3
Job			
	Working	277	43.7
	Construction	332	52.4
	Student	22	3.5
	Retired	3	0.5
Diagnosed with psychiatric illness			
	Yes	5	0.8
	No	629	99.2

**Table 2 healthcare-12-01616-t002:** Prevalence of eating disorders and inappropriate eating behaviors in the sample, and the significant association with the demographic characteristics.

	*n*	%	Overweight before Pregnancy	Pregnancy at 3rd Trimester	School Level	Working
EDE-Q global score ≥4	10	1.60				
Subscale restraint ≥4	28	4.40	*p* = 0.000			
Subscale eating concern ≥4	6	0.90	*p* = 0.000			
Subscale shape concern ≥4	49	7.70				
Subscale weight concern ≥4	35	5.50	*p* = 0.040			
Inappropriate behaviors						
Binge eating	119	18.80	*p* = 0.045	*p* = 0.041		*p* = 0.031
Self-induced vomiting	91	14.40				*p* = 0.030
Laxative misuse	20	3.20	*p* = 0.020		*p* = 0.027	
Excessive exercise	11	1.70	*p* = 0.043			

Footnote: The Eating Disorder Examination Questionnaire (EDE-Q) is a self-report measure used to assess the range and severity of eating disorder behaviors and attitudes. Scores are derived from responses to questions about eating habits, shape, weight, and eating concerns over the past 28 days. Each item is scored on a 7-point Likert scale, ranging from 0 (no days) to 6 (every day). Higher scores indicate more severe eating disorder psychopathology. The EDE-Q generates four subscale scores: restraint, eating concern, shape concern, and weight concern, as well as a global score, which is the average of the four subscales. Scores of 4 and above are considered clinically significant, indicating the presence of eating disorder symptoms.

## Data Availability

The data used for this study are available and will be shared by the corresponding author upon request.

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
