# Peer review of "Eating Behaviors during Pregnancy: A Cross-Sectional Study"

_healthcare, 2024, doi:10.3390/healthcare12161616_

Round 1

Reviewer 1 Report

Comments and Suggestions for Authors

Eating Behaviors during Pregnancy: A Cross-Sectional Study

The authors conducted a study of the eating habits of women in the second and third trimesters of pregnancy who had consulted primary care physicians in Muscat, Oman. The authors used the Eating Disorder Examination Questionnaire (EDE-Q), which had been translated into Arabic. They reported that 1.6% of participants could potentially be diagnosed as having an eating disorder.

The study is interesting, well written, and presents relevant information in each of the sections of the manuscript. Before it is accepted for publication, the following questions must be addressed:

Line 13-14. “We aim to understand how eating disorders affect pregnant women and inform more effective interventions”. The second part of this stated objective is not reflected in the manuscript.

References in the text: Please place the period after each reference.

In the introduction section, please add the main metabolic (not just behavioral) disorders related to these eating disorders so that they relate better to the consequences mentioned in lines 51-56.

Please explain in one or two lines how energy needs increase during the second and third trimesters of pregnancy, and why an eating disorder can compromise the health of the fetus.

Please match the objectives of the study stated in the summary of the manuscript with those written at the end of the introduction section.

Line 72. Why do the authors carry out a study only from January to April 2022? On what basis did they decide only that short duration of time? Do the authors not consider that climate, seasonality, and food availability are variables that can differ throughout the year?

Line 80. If the inclusion criteria indicate pregnant women ≥ 18 years of age, in the second and third trimester of pregnancy, it is not an exclusion criterion that they are in the first trimester of pregnancy. Please modify the wording.

Line 82. Was the translation of the questionnaire into Arabic validated? If so, please include the validation test values. The translated instrument should be validated to guarantee reliability of the results.

Lines 163-174. Can the authors include whether there are differences in the DSM-IV and DSM-5-TR criteria to contribute to the discussion?

Please indicate in your manuscript whether the pregnant women who were identified as possibly having an eating disorder were offered help or had any type of feedback in the study.

Reviewer 2 Report

Comments and Suggestions for Authors

Thank you for the opportunity to review this work. The authors have aimed to identify significant sociodemographic factors associated with eating disorders in Oman. The findings reported a high prevalence of eating disorders in pregnancy, women with lower education reported a higher use of laxative, and working women were more likely to exhibit binge eating behavior and self-induced vomiting.

It is well known to follow a healthy and balanced diet during pregnancy but at the same time it is also important to bring out the other issues which are experienced by pregnant women which impact their diet. It was very interesting that the authors looked into eating disorders experience by pregnant women. Clarifying the below mentioned details will improve the readability of the manuscript.

Comments:

·       Overall suggestion: the authors mention in lines 154-155 that they did not perform a bivariate and multivariate analysis. Could the authors explain why these analyses were not done? Given the data it would be helpful if the authors fit a multivariate model to further identify the direction of association. Is this something planned to include in future work?

·       Line 76: Please mention what prevalence is being described here.

·       Why were women in their first trimester excluded? Given that a lot of food changes happen early on in pregnancy, it would be helpful to look into this phase of pregnancy.

·       Why was the study limited to Omani citizens?

·       Line 98: what does OMR refer to and add to the list of abbreviations?

·        Table 1: please add units of measurement for BMI; under level of education please correct the spelling of college; please include parity or gravidity or both results if available.

·       Table 2: Please add a footnote which describes how to interpret the EDE-Q score.

·       Please add some strengths of the study.

·       Add a statement about generalizability in the discussion.

·       Line 231: Please add explanations about why all cases of eating occurred in the third trimester.

Comments on the Quality of English Language

Minor editing of English language needed.

Author Response

Kindly see attachment

Round 2

Reviewer 1 Report

Comments and Suggestions for Authors

Most of my observations/recommendations were satisfactorily addressed by the authors. Therefore, I consider that the article is publishable in its current form.

Reviewer 2 Report

Comments and Suggestions for Authors

All previous concerns have been addressed.